# Heparin Specifically Interacts with Basic BBXB Motifs of the Chemokine CCL21 to Define CCR7 Signaling

**DOI:** 10.3390/ijms24021670

**Published:** 2023-01-14

**Authors:** Marc Artinger, Oliver J. Gerken, Daniel F. Legler

**Affiliations:** 1Biotechnology Institute Thurgau (BITg), University of Konstanz, Unterseestrasse 47, 8280 Kreuzlingen, Switzerland; 2Graduate School for Cellular and Biomedical Sciences, University of Bern, 3012 Bern, Switzerland; 3Faculty of Biology, University of Konstanz, Universitätsstraße 10, 78464 Konstanz, Germany; 4Theodor Kocher Institute, University of Bern, Freiestrasse 1, 3012 Bern, Switzerland

**Keywords:** CCR7, CCL19, CCL21, biased signaling, GAG, heparin, BBXB motif

## Abstract

Chemokines are critically involved in controlling directed leukocyte migration. Spatiotemporal secretion together with local retention processes establish and maintain local chemokine gradients that guide directional cell migration. Extracellular matrix proteins, particularly glycosaminoglycans (GAGs), locally retain chemokines through electrochemical interactions. The two chemokines CCL19 and CCL21 guide CCR7-expressing leukocytes, such as antigen-bearing dendritic cells and T lymphocytes, to draining lymph nodes to initiate adaptive immune responses. CCL21—in contrast to CCL19—is characterized by a unique extended C-terminus composed of highly charged residues to facilitate interactions with GAGs. Notably, both chemokines can trigger common, but also ligand-biased signaling through the same receptor. The underlying molecular mechanism of ligand-biased CCR7 signaling is poorly understood. Using a series of naturally occurring chemokine variants in combination with newly designed site-specific chemokine mutants, we herein assessed CCR7 signaling, as well as GAG interactions. We demonstrate that the charged chemokine C-terminus does not fully confer CCL21-biased CCR7 signaling. Besides the positively charged C-terminus, CCL21 also possesses specific BBXB motifs comprising basic amino acids. We show that CCL21 variants where individual BBXB motifs are mutated retain their capability to trigger G-protein-dependent CCR7 signaling, but lose their ability to interact with heparin. Moreover, we show that heparin specifically interacts with CCL21, but not with CCL19, and thereby competes with ligand-binding to CCR7 and prevents signaling. Hence, we provide evidence that soluble heparin, but not the other GAGs, complexes with CCL21 to define CCR7 signaling in a ligand-dependent manner.

## 1. Introduction

Chemokines are small chemotactic cytokines involved in orchestrating the directed migration of immune cells and thus play a central role in the development and homeostasis of the immune system [1]. As chemokines bear overall cationic characteristics, they are prone to interact with negatively charged glycosaminoglycans (GAGs). GAGs thereby confine chemokines on and near cell surfaces and the extracellular matrix to provide local guidance cues [2]. GAGs are classified into four different groups: heparin or heparan sulfate, chondroitin or dermatan sulfate, keratan sulfate and hyaluronic acid, a non-sulfated GAG non-covalently linked to membrane proteins [3,4]. Of note, chemokine-GAG interactions are mainly based on van-der-Waals bonds and hydrophobic, electrostatic interactions between the carbohydrate backbone of glycoprotein-linked glycans and the positively charged residues of the chemokines [2,4,5]. Based on those electrochemical interactions, chemotactic and haptotactic chemokine gradients are established and maintained in vivo, controlling the directed migration and homing of leukocytes, such as dendritic cells that migrate from peripheral tissues to draining lymph nodes in response to gradients of the chemokine CCL21 [6]. Interfering with the chemokine-GAG interaction, for example by the elimination of extracellular matrix (ECM) attached heparan sulfates on the endothelium of lymph nodes, was shown to abrogate correct CCL21 presentation and subsequently efficient lymphocyte homing [7].

Chemokines are recognized by their cognate chemokine receptors, which belong to the class A of G-protein coupled receptors [8]. This receptor class couples to heterotrimeric G_i_ proteins and stimulates the mobilization of calcium ions from intracellular stores and the production of cytosolic cAMP and other second messengers resulting in actin rearrangements and directed cellular migration [8,9,10]. One of these chemokine receptors is the CC motif chemokine receptor 7 (CCR7), which together with its cognate chemokine ligands CCL19 and CCL21, mediates the homing of specialized leukocyte subsets to secondary lymphoid organs and thereby balances adaptive immunity and tolerance [11]. Notably, CCL19 and CCL21 trigger common CCR7 signaling routes, involving G protein activation or phosphorylation of extracellular-signaling regulated kinases ERK1/2, but these two chemokine ligands differ in their ability to induce receptor internalization [1,12,13,14,15]. The critical role of the CCR7-CCL19/CCL21 axis in immunity is manifested by significantly reduced numbers of naïve T cells and dendritic cells that home to lymph nodes in *plt/plt* (paucity in lymph node T cells) mice, which lack CCL21 and CCL19 expression in secondary lymphoid organs [11,16,17]. In addition to leukocytes, cancer cells of various subtypes upregulate the surface expression of CCR7, a circumstance that correlates with a higher level of lymph node metastasis and a worse overall prognosis [18,19,20,21,22]. Of note, the enhanced lymph node metastasis was shown to be connected to the increased binding ability of CCL21 to extracellularly presented GAGs on lymphatic vessels [23].

CCL19 and CCL21 share only 32% sequence similarity and show major differences in their protein N-termini as well as in the characteristic chemokine core structure [24,25,26,27]. The chemokine core structure usually covers about 70 amino acids and is composed of a three-stranded antiparallel β-sheet connected via the 30s and the 40s loop, as well as a C-terminal α-helix [28]. The most striking difference between CCL19 and CCL21 is the unique, highly charged C-terminal extension of CCL21 comprising several basic amino acids, a feature that is not present in most chemokines [24,25]. Despite the unstructured nature, the C-terminus of CCL21 also includes two additional cysteine (C) residues forming an additional third disulfide bond between C80 and C99 [29]. After proteolytic cleavage by plasmin, which is secreted, e.g. by mature dendritic cells, CCL21 also exists in a naturally occurring truncated form lacking the polybasic stretch at its C-terminus rendering the chemokine more soluble and alike to CCL19 [30,31]. This C-terminal CCL21 truncation was shown to affect chemokine-mediated CCR7 signaling and chemotaxis [26,27,32]. Interesting, but partially conflicting signaling capabilities are reported on an artificial chimeric chemokine termed CCL19chim, in which the native CCL19 is extended with the C-terminus of CCL21. A decreased chemotactic activity, but increased GAG binding on human dendritic cells was shown for CCL19chim as comparable to CCL21, while CCL19chim and CCL19 possessed similar activity for intracellular calcium mobilization and receptor endocytosis [26,33,34]. The properties shared by CCL19chim and the native form of CCL21 regarding GAG binding might also derive from specific clusters of basic amino acids located within the C-terminus, which were shown to be important for the binding to heparan sulfate [3,35,36]. These polar stretches consist of several adjacent basic amino acids spaced by any amino acid and correspond to so-called BBXB or BBBXXB consensus sequences (B: basic amino acid; X: any amino acid). Such BBXB motifs are mainly involved in the electrochemical interaction of proteins and glycosaminoglycans [37]. Notably, CCL21 owns three distinct BBXB motifs [34].

Strikingly, by sharing the same receptor, CCL19 and CCL21 trigger both common but also distinct intracellular signaling pathways [1]. Particularly, CCR7 stimulation by both ligands results in G protein activation and downstream signaling, whereas receptor internalization is induced predominantly upon CCL19 triggering [14]. A recent study identified the chemokine core structure rather than its N-terminus to account for biased signaling [38]. However, little is known about whether and how chemokine-GAG interactions modulate (ligand-biased) signaling. In the present study, we assessed whether alterations in the predicted glycosaminoglycan interacting regions of CCL21 would lead to changes in biased signaling and how heparin binding would affect CCR7 signaling. A better understanding of the dynamics between the CCR7-CCL19/CCL21 axis and surface-presented GAG structures represents a valuable strategy in therapeutic drug development.

## 2. Results

### 2.1. Charged C-Terminal Residues of CCL21 Only Partially Confer Ligand-Biased CCR7 Signaling

CCL19 and CCL21, through binding to their cognate receptor CCR7, have been shown to trigger common as well as ligand-biased signaling, for which the molecular mechanism is faintly understood. In addition to the native chemokines CCL19 and CCL21, the latter also exists as a naturally occurring truncated variant lacking the prolonged and highly charged C-terminus, known as CCL21trunc (Figure 1a), which shares some characteristics with CCL19 [1]. To gain insights into the functional role of the CCL21 tail, we fused it C-terminally to CCL19 and referred to it as CCL19chim (Figure 1a). Previous studies reported that CCL19 is more potent in recruiting βarrestins to CCR7 than CCL21 [38]. Here, we used a split-luciferase-based assay to systematically measure the chemokine-mediated recruitment of βarrestin2 to CCR7 in a ligand-concentration-dependent manner (Figure 1b). Indeed, CCL19 owned the highest potency in recruiting βarrestin2 to CCR7 (EC_50_ 38 nM). In comparison, CCL21 was about seven times less potent (EC_50_ 276 nM). Notably, CCL21trunc (EC_50_ 242 nM) behaved similarly to CCL21 (Figure 1b), whereas transferring the CCL21 C terminus to CCL19 in the CCL19chim partially reduced its potency in recruiting βarrestin2 (EC_50_ 122 nM) (Figure 1b).

Next, we assessed the chemokine-driven recruitment of the G protein surrogate MiniG_αi_ to CCR7, which serves as a readout for receptor activation [39]. Again, CCL19 was more potent than CCL21 in recruiting MiniG_αi_ (Figure 1c), which is consistent with the reported chemokine-mediated phosphatidylinositol turnover used as a readout of G_i_ signaling [38]. Interestingly, CCL19, CCL19chim, CCL21, and CCL21trunc showed similar differences in their abilities to recruit βarrestin2 and MiniG_αi_ (Figure 1c). However, both native chemokines, CCL19 (EC_50_ 19 nM) and CCL21 (EC_50_ 38 nM), efficiently mobilized Ca^2+^ from intracellular stores, which is downstream of G protein activation (Figure 1d). Assessing all four chemokine variants at a fixed nanomolar concentration revealed comparable abilities in Ca^2+^ mobilization (Figure 1e). These data suggest that the C-terminus of CCL21 alone is not the key driver for ligand-biased βarrestin recruitment, but nonetheless modulates CCR7 signaling.

Besides positively charged amino acids located at the C-termini of chemokines, specific motifs consisting of basic amino acids, e.g., BBXB or BBBXXB, are known to mediate glycosaminoglycan binding but might be also involved in receptor interaction [2]. Notably, CCL19 and CCL21 possess one and three of those motifs, respectively. We, therefore, generated chemokine variants where the basic amino acids within these motifs were mutated to alanines. Unfortunately and for an unknown reason, the ^74^KRR^76^ to ^74^AAA^76^ CCL19 mutant could not be produced in our bacterial host system. However, we successfully expressed and purified three different BBXB mutants for CCL21. Namely, CCL21 M1, refers to a CCL21 variant where the first BBXB motif ^44^RKR^46^ located within the chemokine protein core was mutated (Figure 2a); CCL21 M2 and CCL21 M3 refer to chemokine variants where the mutated BBXB motifs ^81^RKDR^84^ and ^91^KKGK^94^ are part of the polar C terminal tail region (Figure 2a).

All three variants of CCL21 with individually mutated BBXB motifs were biologically active and able to recruit MiniG_αi_ to CCR7 (Figure 2b). Interestingly, the CCL21 variants M2 and M3 where the mutated BBXB motif is located within the C-terminus of CCL21 were as efficient in MiniG_αi_ recruitment to CCR7 as wild-type CCL21 was (CCL21 EC_50_ 389 nM, CCL21 M2 EC_50_ 440 nM, CCL21 M3 EC_50_ 406 nM) (Figure 2b). In contrast, CCL21 M1 where the BBXB motif in the chemokine core domain is mutated was less efficient in recruiting MiniG_αi_ to CCR7 (CCL21 M1 EC_50_ 938 nM) (Figure 2b). Again, no significant difference in their ability to mobilize intracellular Ca^2+^ in response to CCR7 triggering was detected for the four CCL21 forms (Figure 2c) suggesting that the presence or absence of positively charged residues on the chemokine surface per se cannot fully explain ligand-biased CCR7 signaling.

### 2.2. Positively Charged Amino Acids at the Surface of CCL21 Mediate Heparin Binding

In general, negatively charged ECM properties are considered key drivers of chemokine retention and hence help shape haptotactic gradients in tissues. Chemokines electrochemically interact with ECM substrates via basic surface amino acids bearing an overall positive charge at physiological pH. Correlating the absolute numbers of basic surface amino acids present in different forms of CCL19 and CCL21 with the computational isoelectric point of the protein revealed a link between the basic amino acid content of the corresponding chemokine and the overall surface charge (Figure 3a,b). In particular, truncated forms of CCL21, namely CCL21trunc81, CCL21trunc79, and CCL21trunc74 that naturally occur from the sequential proteolytic cleavage by plasmin and consequently lack the two BBXB motifs in the C terminal region of the chemokine display the lowest overall net surface charge at a neutral pH (Figure 3b). Consistent with this, individual and combinatory alternations in different BBXB motifs of CCL21 possess intermediate net surface charges at a neutral pH.

To experimentally assess the ECM interaction abilities of different CCR7 ligand variants, sepharose-conjugated heparin was used as a surrogate system allowing chemokine trapping and binding, followed by elution using a chaotropic buffer with gradually increasing concentrations of sodium chloride (Figure 3c). In line with the calculated chemokine variant-specific PI values, the binding strength of CCL21 to heparin was higher than that for the less positively charged CCL19 (Figure 3d). This difference in interaction can be attributed to the CCL21′s C-terminus, as transfer thereof led to increased heparin-binding for CCL19chim, whereas CCL21trunc showed reduced binding and was comparable to that of CCL19 (Figure 3d). Interestingly, all the single BBXB motif CCL21 mutants showed partially decreased electrochemical interactions with heparin, suggesting that the individual BBXB motifs of CCL21 conjointly contribute to heparin binding. Attempts to experimentally support this notion failed as CCL21 variants where two or three BBXB motifs were simultaneously mutated could not be expressed and purified.

### 2.3. Heparin Interacts with CCL21′s C-terminus and Negatively Regulates CCR7 Signaling

As CCL21 strongly interacted with heparin (Figure 3), we wondered whether the addition of exogenous free heparin would affect chemokine binding to CCR7 and signal transduction. To assess this, we first performed a chemokine binding competition assay, wherein binding of a fixed concentration (25 nM) of fluorescently labeled CCL21-S6^Dy649P1^ to 300-19 cells stably expressing human CCR7 was measured while titrating in heparin (Figure 4a). As expected, CCL21-S6^Dy649P1^ binding to CCR7 was higher at 37 °C than at 4 °C where the plasma membrane is rigid. Notably, the addition of graded concentrations of free heparin outcompeted CCL21-S6^Dy649P1^ binding to CCR7 with an IC_50_ of 153 ng/mL at 4 °C and an IC_50_ of 178 ng/mL at 37 °C, respectively (Figure 4a). Next, we determined the effect of adding exogenous heparin on chemokine-driven MiniG_αi_ recruitment to CCR7. We found that CCL21-mediated MiniG_αi_ recruitment to CCR7 was inhibited in a concentration-dependent manner upon titrating in free heparin with an IC_50_ of 43 µg/mL (Figure 4b). By contrast, CCL21trunc-driven MiniG_αi_ recruitment virtually remained unaffected by the addition of heparin and showed inhibitory effects only at very high heparin concentrations, indicating that the C-terminus of CCL21 is key for heparin binding. Correspondingly, transferring the C terminus of CCL21 to CCL19 led to a heparin concentration-dependent inhibition of MiniG_αI_ recruitment to CCR7 by CCL19chim, but not for native CCL19 (Figure 4c). These data corroborate the notion that the interaction of heparin with CCL21 relies on its highly polar C terminus and that the interaction of heparin with the chemokine limits chemokine binding and signaling through its cognate receptor.

We wondered whether other glycosaminoglycan structures known to interact with chemokines [33,40] would also interfere with chemokine binding and signaling through CCR7. Therefore, we tested chondroitin sulfate A (CS A) and sialic acid (Sia) in comparison to heparin (Hep) to modulate CCL19 and CCL21 driven βarrestin2 recruitment to CCR7 and ligand-mediated mobilization of cytosolic free Ca^2+^. Notably, none of the tested ECM glycan structures affected CCL19-mediated CCR7 signaling (Figure 5a,b). Interestingly, βarrestin2 recruitment to CCR7 and Ca^2+^ mobilization by CCL21 was specifically inhibited in the presence of heparin, but not by chondroitin sulfate A or sialic acid (Figure 5a,b).

In summary, here we demonstrated that individual charged motifs within the chemokine core domain and the C-terminus of CCL21 are not the main drivers for the differences in ligand-biased CCR7 signaling, but that the BBXB motifs account for electrochemical interactions of the chemokine with heparin. Moreover, we provide evidence that soluble heparin, but not the other GAGs, is able to form a complex with CCL21 to define and interfere with CCR7 signaling in a ligand-dependent manner.

## 3. Discussion

Ligand-biased signaling of G-protein coupled receptors is an important and intensely studied concept. CCR7 and its two ligands gained a lot of attention due to its ligand-biased signaling capacities, although the underlying molecular mechanism remains poorly understood. In this study, we demonstrate that the unique and highly charged C terminus of CCL21 is not decisive for ligand-biased signaling. However, intramolecular BBXB motifs located either at the C-terminus or within the chemokine core domain of CCL21 account for the interaction with the ECM protein heparin. Notably, individual BBXB motifs of CCL21 conjointly contribute to heparin-binding with the strongest contribution of the chemokine’s C-terminus. Consequently, fusing the CCL21′s tail to CCL19 fully conferred heparin binding to CCL19chim, but only marginally affected ligand-biased CCR7 signaling. Interestingly, heparin addition to chemokine-receptor binding and signaling assays indicates a ligand-specific formation of a complex with CCL21, but not with CCL19, that defines and limits CCR7 signaling in a ligand-dependent manner. Our results thus extend previous studies assessing the interaction of CCR7 ligands with different ECM proteins [33,41,42]. In addition, our study sheds light on the role of BBXB motifs and the charged C-terminus of CCL21 in receptor activation and ECM interaction.

Interestingly, recent studies described differences in ligand-biased CCR7 signaling among different cell types. Whereas in primary leukocytes, such as human matured dendritic cells and peripheral blood mononuclear cells, chemokine-mediated Ca^2+^ mobilization was substantially more pronounced for CCL21 compared to CCL21trunc; the latter was reported to be more potent in the commercially available adherent CHEM-1 cell line frequently used for high throughput assessing of GPCR functions [26,27,42]. Discrepant findings were also reported for CCL21 and CCL21trunc in terms of their abilities to recruit βarrestin: one study found comparable efficiencies for the two CCL21 forms [27], whereas the other study observed a higher potency for CCL21trunc compared to CCL21 [43]. Furthermore, transferring the C-terminus of CCL21 to the tail of CCL19 in CCL19chim was found to more efficiently bind to dendritic cells than CCL19 without affecting the chemotactic response, whereas ligand-mediated CCR7 internalization was diminished [26]. Our data presented herein are in line with a previous study [32] and reveal that CCL21trunc does not simply behave like CCL19 but owns unique biological activities. The notion that tissue bias can contribute to modulating CCR7 signaling [33] adds an extra layer of complexity in understanding (ligand-) biased signaling. This likely explains the differences in potency and efficacy of Ca^2+^ mobilization capacities induced by CCL19, CCL21, and variants thereof in different cell types. Hence, signaling bias must be seen and interpreted in a cellular context.

It is well established that specific basic amino acid stretches in chemokines are associated with the probability to interact with extracellularly presented GAG structures [2]. For instance, specific BBXB motifs were shown to be involved in heparan sulfate binding of CXCL12 [44] and of chemokines with prolonged C-termini such as CXCL9 [37]. Here, we report that CCL21 harbors three BBXB motifs and that only mutations in either BBXB motif of the chemokine core domain led to a minimal decrease in signaling efficiency. Similar effects have been observed for CCL5, for which a BBXB variant within the core domain showed a decreased chemotactic activity for monocytes [45]. Among several investigated GAGs, mainly heparin and chondroitin sulfate B exhibited the most prominent modulating effect on chemokine signaling, with a stronger effect of soluble heparin over the more physiological sulfated form heparan sulfate [41,44]. Interestingly, free heparin or heparan sulfate enhances neutrophil responses towards CXCL8, whereas free forms of heparin diminished cellular responses induced by CCR7 ligands [3,41,42]. Our chemokine competition assay presented herein supports a previous model suggesting the binding of free heparin to CCL21 releases the chemokine from an autoinhibitory monomeric state, and therefore, enhances chemokine-related signaling [43]. However, instead of boosting CCL21 functions, the binding of free heparin to the chemokine in our hands limited its accessibility to bind to CCR7 and consequently prevented CCL21-mediated CCR7 signaling. Of note, a recent study presented that the addition of free C terminal peptide fragments of CCL21 boosted CCL19- and CCL21-mediated signaling, presumably by binding to and saturating surface exposed GAG structures and thereby facilitating chemokine binding to its receptor [46].

In conclusion, further studies assessing the CCR7-CCL19/CCL21 axis are needed to fully understand the complexity of how individual chemokines interact with different GAGs in concert with triggering cognate receptor signaling pathways. Notably, a CCL5 variant with a mutated BBXB motif and decreased GAG binding capacities efficiently inhibited HIV infection through CCR5 in vitro without affecting CCR5-mediated monocyte migration [45]. In the light of potentially interfering with CCR7-mediated metastasis formation without perturbing CCR7-guided adaptive immune cell homing, new insights on cell-biased and ligand-biased CCR7 signaling is highly desired.

## 4. Material and Methods

### 4.1. Plasmids

Cloning of His_6_-SUMO-hCCL19chim and His_6_-SUMO-hCCL21trunc79 was described elsewhere [47,48]. Furthermore, BBXB motif variants of CCL21, namely M1 (aa44–46, RKR), M2 (aa81–84, RKDR) and M3 (aa91–94, KKGK), were exchanged for alanine via site-directed mutagenesis with the following primer pairs based on His6-SUMO-hCCL21 [32]: for 5′CTA TCC TGT TCT TGC CCG CAG CAG CAT CTC AGG CAG AGC TAT G, rev 5′CAT AGC TCT GCC TGA GAT GCT GCT GCG GGC AAG AAC AGG ATA G for M1, for 5-′CCA CAG AAA CCA GCC CAG GGC TGC GCA GCA GAC GCA GGG GCC TCC AAG ACT GGC AAG AAA G and rev 5-′CTT TCT TGC CAG TCT TGG AGG CCC CTG CGT CTG CTG CGC AGC CCT GGG CTG GTT TCT GTG G for M2 and for 5′CTC CAA GAC TGG CGC AGC AGG AGC AGG CTC CAA AGG CTG and rev 5′CAG CCT TTG GAG CCT GCT CCT GCT GCG CCA GTC TTG GAG for M3, respectively.

For split luciferase experiments, pcDNA3 hCCR7-NLuc11 (aa sequence GWRLCERILAG) was amplified with flanking EcoRI and XbaI cutting sites and a (GGGGS)_3_ linker from pcDNA3 hCCR7-EGFP [14] using the primer pair for 5′CGA AAT TAA TAC GAC TCA CTA TAG GGA GAC CC and rev 5′GCT CCT CGC CCT TGC TCA CTC TAG ACT AGC CCG CCA GAA TGC GTT CGC ACA GCC GCC AGC CGC TAC CGC CAC CGC CGG A. After amplification, CCR7-NLuc11 was ligated into the corresponding cutting sites of the pcDNA3 backbone. For the split luciferase counterparts, pcDNA3 NLuc158-MiniG_αi_ was amplified from pBit1.1 lgBit-MiniG_αi_ [39] with the primers for 5′GCG ACC CGC TTA AAA GAA TTC TTG GCA ATC CGG TAC and rev 5′CGG CCG CCC CGA CTC TAG AAG ATC TGC TAG CTT AGA CTG AAT TTG GCG CTT GTT AGA AA creating EcoRI and XbaI cutting sites which were then ligated into a pcDNA3 backbone. In addition, the primer pair for 5′CAA CGG AGT GAC CTG ATC ACG GCT GTG CGA ACG CAT TC and rev 5′GCG TTC GCA CAG CCG TGA TCA GGT CAC TCC GTT GAT GGT TAC was used to generate pcDNA3 βarrestin2i1-NLuc158 (aa1-158) from βarrestin2i1-NLuc [49] and was ligated again in the EcoRI, XbaI cutting sites of a pcDNA3 backbone.

### 4.2. Chemokine Purification and Fluorescent Labeling

Human chemokines (CCL19, CCL19chim, CCL21, CCL21-S6^Dy649P1^, CCL21trunc79 and the corresponding BBXB motifs) were purified and labeled as previously described [48].

### 4.3. Cell Culture

Human HeLa cells were cultivated in DMEM (Pan Biotech, Aidenbach, Switzerland) supplemented with 10% fetal calf serum (FCS; Lonza, Basel, Switzerland) and 1% penicillin/streptomycin (P/S; Pan Biotech). Murine 300-19 pre B cells stably expressing human CCR7 [50,51] were maintained in RPMI-1640 medium (Pan-Biotech) supplemented with 10% FCS, 1% P/S, 1% non-essential amino acids (Biowest, Riverside, MO, USA) and 0.1% β-mercaptoethanol (Gibco, ThermoFisher, Carlsbad, CA, USA). Cells were cultured at 37 °C, 5% CO_2_ and 95% humidity.

### 4.4. Transient Transfection

HeLa cells were transiently transfected with 10 µg total plasmid DNA coding for lgBit-MiniG_αi_ or βarrestin2-NLuc158 and CCR7-NLuc11 in a 1:1 ratio. Therefore, cells were electroporated using the Neon Transfection System (ThermoFisher) according to the manufacturer’s protocol and 100 µL tips before seeding transfected cells in 6-well plates containing DMEM supplemented with 20% FCS. Allowing receptor surface expression, cells were harvested after 36–48 h for subsequent experiments.

### 4.5. MiniG_αi_ and βarrestin2 Recruitment to CCR7

Chemokine-mediated recruitment of MiniG_αi_ or βarrestin2 to CCR7 was investigated by split luciferase assays [52], which rely on the complementation of two NanoLuciferase subunits upon close proximity of the two proteins of interest. In brief, transiently transfected HeLa cells were washed with PBS supplemented with 5 mM glucose (PBS-G), detached using PBS-G supplemented with 0.5 mM EDTA and collected with DMEM. Cells were then washed and resuspended in PBS-G and loaded with 5 µM of the luciferase substrate CoelenthrazineH (Biosynth, Staad, Switzerland). After distribution into a 96-well half-area plate, baseline luminescence was measured for 10 min (384–440 nm, 500 ms integration time) before stimulation with the indicated concentrations of chemokine. Luminescence was further recorded over a period of 30 min on a Spark M10 microplate reader (Tecan, Männedorf, Switzerland).

For characterizing the effect of different glycosaminoglycans on chemokine signaling, cells were simultaneously stimulated with different concentrations of chondroitin sulfate A (Roth, Arlesheim, Switzerland), heparin (Roth), N-acetyl neuraminic acid (Roth) or BSA (Roth).

### 4.6. Mobilization of Intracellular Calcium

To investigate changes in intracellular Ca^2+^ levels upon chemokine stimulation, 300-19 cells stably expressing hCCR7 were harvested and resuspended in calcium flux buffer (145 mM NaCl, 5 mM KCl, 1 mM MgCl_2_, 1 mM CaCl_2_, 1 mM sodium phosphate, 5 mM Hepes, pH 7.5) at a concentration of 1 × 10^6^ cells/mL. Then, cells were loaded with 4 µM Fluo3-AM (ThermoFisher) at 37 °C for 30 min. After extensive washing, cells were adjusted to 3 × 10^6^ cells/mL and Fluo3-associated fluorescence was measured over time on an LSR Fortessa flow cytometer (BD Biosciences, San Jose, CA, USA). Following the 30 s baseline measurement, cells were stimulated with indicated concentrations of chemokine to determine the EC_50_ values and to characterize chemokine variant functionality. Acquired data were analyzed with the FACS Diva Software and FlowJo10 (BD Biosciences). Ca^2+^ mobilization data were normalized to the maximal Fluo3 fluorescence achieved by stimulation of the cells with 1 µM ionomycin (Sigma, St. Louis, MO, USA).

### 4.7. Heparin Binding Assays

Chemokine binding to a heparin matrix was performed on an ÄKTA Pure 25M2 System equipped with an external sample pump (GE Healthcare, Chicago, IL, USA) using a HiTrap Heparin HP (Cytiva Life Sciences, Glattbrugg, Switzerland) column. Lyophilized chemokines were reconstituted in sterile-filtered PBS (Fisher BioReagents, Reinach, Switzerland), and 50 µg of chemokine per run were automatically loaded onto the HiTrap Heparin HP column using a sample loop. After washing, column elution was performed with gradually increasing concentrations of PBS containing 2 M NaCl (Sigma). Chemokine binding strength was determined by comparing the conductivity during the peak elution at OD_280nm_. Analyzed peaks were normalized to their maximal absorption at 280 nm.

### 4.8. Competition for Chemokine Binding to CCR7 and by Heparin

The competition of heparin in the binding of CCL21 to CCR7 was assessed by flow cytometry. Therefore, 2 × 10^5^ 300-19 cells stably expressing hCCR7 were stimulated in a 96-well format with 25 nM CCL21-S6^Dy649P1^ and increasing concentrations of unfractionated heparin at 4 °C or 37 °C for 1 h. After washing with calcium flux buffer, cells were analyzed for chemokine-associated fluorescence by flow cytometry and the signal of CCL21-S6^Dy649P1^ for non-inhibited cells at 37 °C was set to 100% as the corresponding control.

## Figures and Tables

**Figure 1 ijms-24-01670-f001:**
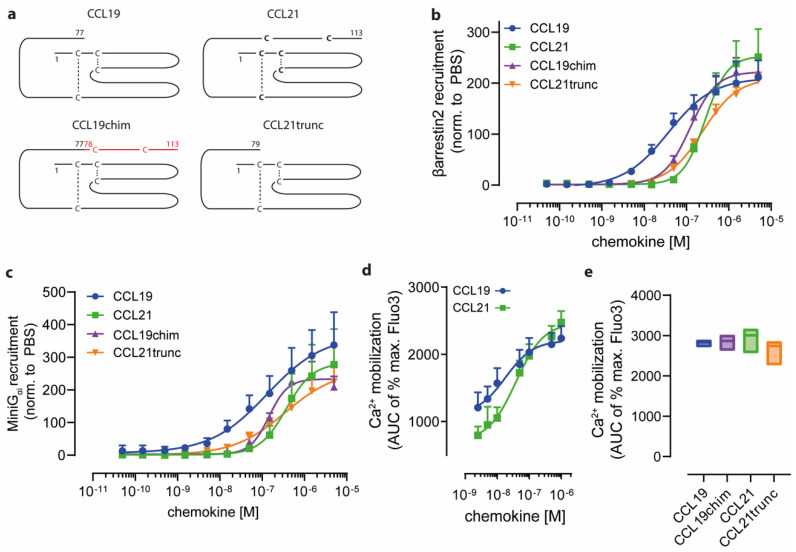
Ligand-biased CCR7 signaling only partially relies on the C terminal tail of CCL21. (**a**) Schematic representation of native CCL19 and CCL21, the naturally occurring C-terminally truncated chemokine variant CCL21trunc and CCL19chim, a CCL19 variant extended by the CCL21′s C terminus (marked in red). Characteristic disulfide bonds (dashed lines) between cysteine (C) residues and the specific chemokine length indicated by the number of the last amino acid are depicted. (**b**) βarrestin2-NLuc158 recruitment to CCR7-NLuc11 upon chemokine stimulation with graded concentrations of each chemokine variant. The area under the curve (AUC) analysis for each chemokine representing the mean ± SD of n = 3 is shown. (**c**) LgBit-MiniG_αi_ is recruited to CCR7-NLuc11 in a concentration-dependent manner for all tested chemokine variants. Normalized mean AUC values ± SD of n = 3 are shown. (**d**) Calcium mobilization in 300-19 pre-B cells expressing CCR7 in response to graded concentrations of native CCL19 and CCL21. Mean AUC values ± SD of three independent experiments that were normalized to the maximal fluorescence achieved with ionomycin are shown. (**e**) Experimental setup as in (**d**). 300-19 cells expressing CCR7 were stimulated with 50 nM of the indicated chemokine. Median AUC values from min. to max. of n = 3 are shown.

**Figure 2 ijms-24-01670-f002:**
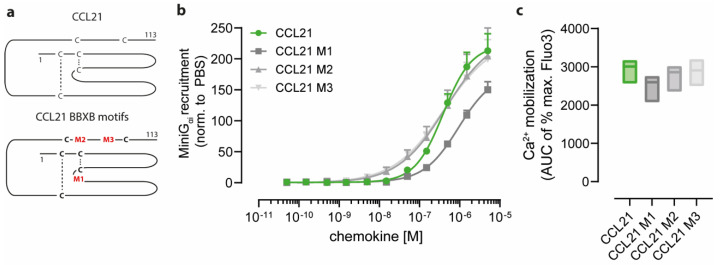
CCL21 variants with individually mutated BBXB motifs remain functional. (**a**) Scheme of CCL21 where the three BBXB motifs are marked in red with M1, M2 and M3. (**b**) Recruitment of lgBit-MiniG_αi_ to CCR7 upon stimulation with graded concentrations of either native CCL21 or its BBXB motif mutants. Normalized mean AUC values ± SD of n = 3 for each chemokine variant are shown. (**c**) Intracellular Ca^2+^ mobilization of 300-19 pre-B cells expressing CCR7 upon stimulation with 50 nM of the indicated CCL21 variant. Median AUC values from min. to max. of three independent experiments that were normalized to the maximal fluorescence induced by ionomycin are shown.

**Figure 3 ijms-24-01670-f003:**
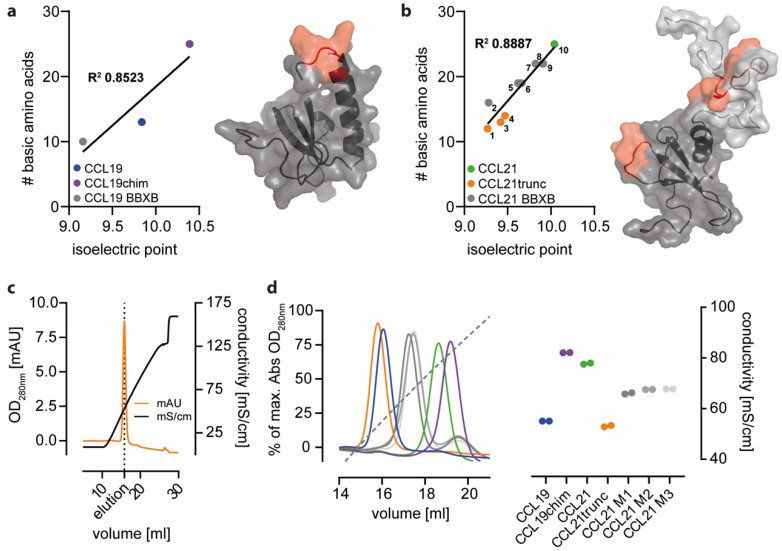
Basic amino acids on the chemokine surface determine heparin binding strength. (**a**/**b**) Correlation of the actual number of basic surface amino acids and the corresponding calculated chemokine isoelectric point of different CCL19 (**a**) and CCL21 (**b**) variants (1: CCL21trunc74, 2: CCL21 M1/2/3, 3: CCL21trunc79, 4: CCL21trunc81, 5: CCL21 M1/3, 6: CCL21 M1/2, 7: CCL21 M1, 8: CCL21 M3, 9: CCL21 M2, 10: CCL21). Positions of basic amino acids (red) within the chemokines are depicted in the corresponding protein surface representation (PDB: CCL19 2MP1; CCL21 2L4N). The polar C-terminus of CCL21 is shown in light gray. (**c**) Heparin binding strength was determined by loading a HiTrap heparin HP column with 50 μg of the indicated chemokine, followed by elution with 2 M sodium chloride. Conductivity (black line) and absorbance at 280 nm (orange line) were used to compare the individual elution profiles. Profile of CCL19chim is shown as an example. (**d**) Elution profiles of different chemokine variants as indicated in (**c**) (left; color code as depiced in the right panel) and the quantitative analysis of two independent experiments (right) showing different chemokine binding capacities to immobilized heparin using the indicated color coding for different chemokine variants.

**Figure 4 ijms-24-01670-f004:**
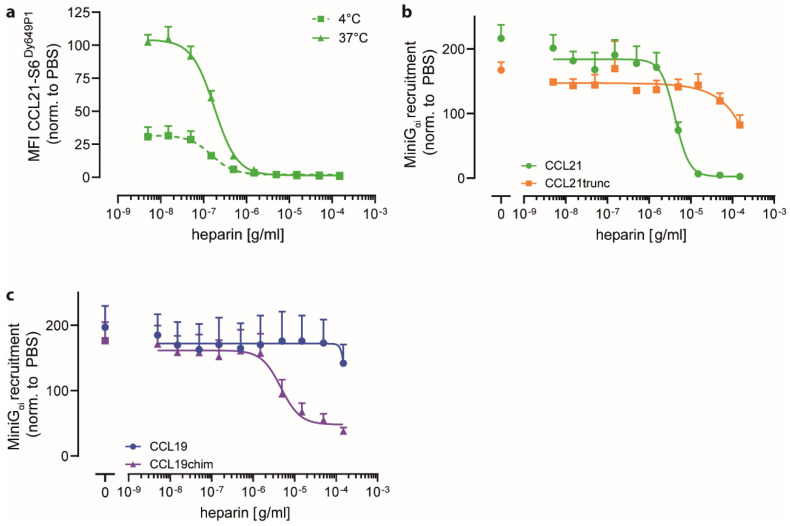
Soluble heparin interacts with CCL21′s C-terminus and limits chemokine-mediated CCR7 signaling. (**a**) 300-19 pre-B cells stably expressing CCR7 were incubated with 25 nM of fluorescently labeled CCL21-S6^Dy649P1^ and increasing concentrations of exogenous heparin for 30 min at 4 °C (dotted line) or 37 °C (solid line). Relative chemokine fluorescence was normalized to cells treated with fluorescent chemokine and PBS. Shown are mean values ± SD of n = 3. (**b**,**c**) Recruitment of lgBit-MiniG_αi_ to CCR7-NLuc11 upon stimulation with 1 μM of either CCL21 or CCL21trunc (**b**) or 1 μM CCL19 and CCL19chim, respectively (**c**), in the presence of increasing concentrations of heparin. Mean AUC values ± SD of three independent experiments.

**Figure 5 ijms-24-01670-f005:**
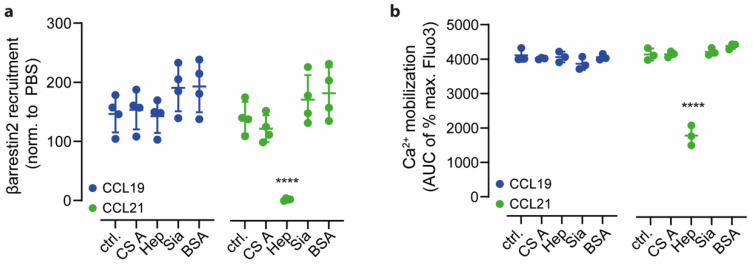
Heparin, but not other glycosaminoglycans, specifically regulate CCL21-mediated signaling. (**a**) Recruitment of βarrestin2-NLuc158 to CCR7-NLuc11 upon stimulation with 1 μM CCL19 (left) or CCL21 (right) in the presence of PBS (ctrl) or 50 μg/mL of either chondroitin sulfate A (CS A), heparin (Hep), sialic acid (Sia) or bovine serum albumin as a negative control (BSA). AUC analysis of mean values ± SD of n = 4 is depicted. (**b**) Intracellular calcium mobilization in response to 50 nM of either CCL19 (left) or CCL21 (right) in the presence or absence of indicated ECM proteins. Mean AUC values ± SD of n = 3, statistical analysis: one-way ANOVA with Dunnett correction. Shapiro–Wilk was used as test for normality. **** *p* ≤ 0.0001.

## Data Availability

Datasets for this study are deposited on Zenodo and are publicly available under a Creative Commons Attribution 4.0 International license, doi: 10.5281/zenodo.6362293.

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
