# Peer review of "Heparin Specifically Interacts with Basic BBXB Motifs of the Chemokine CCL21 to Define CCR7 Signaling"

_ijms, 2023, doi:10.3390/ijms24021670_

Round 1

Reviewer 1 Report

This paper introduced a significant approach for understanding  the underlying molecular 19 mechanism of ligand-biased CCR7 signaling. In my opinion the work is offering new insights for the researchers in the same field.

Author Response

We thank reviewer 1 for the appraisal of our work!

Reviewer 2 Report

ijms-2134854

Heparin specifically interacts with basic BBXB motifs of the chemokine CCL21 to define CCR7 signaling

This article focuses on the molecular mechanisms determining ligand-biased CCR7-dependent signaling induced by CCL19 and CCL21, respectively, and the impact of interactions with glycosaminoglycans (GAGs; esp. heparin) on the modulation of CCR7 signaling. The authors report that both the highly charged CCL21 C-terminus as well as the presence of three basic amino acid-rich motifs only partially contributes to CCL21-biased CCR7 signaling. The higher positive charge of CCL21 in comparison to CCL19, however, results in a stronger association with heparin. Moreover, in contrast to other GAGs, heparin specifically competes with CCR7 for CCL21 binding thus affecting CCL21-induced CCR7 signaling.

The study is well designed, carried out properly, and technically sound. The manuscript is excellently written and the presentation of the data is straightforward and clear. The results are comprehensible and conclusive. Thus, there are only a few points/questions requiring the authors’ consideration.

1. Results: I assume that unfractionated (high molecular weight) heparin was used for the experiments. To assess whether the length of the GAGs plays a role, it should be tested whether fractionated (low molecular weight) heparin has equivalent effects.

2. In Figure 3b, it should be indicated which truncated versions and BBXB variants of CCL21 are represented by the respective dots.

3. Since the increasing doses of heparin show at least a mild effect on CCL21trunc-driven MiniGαi recruitment in high concentrations in Figure 4b, I suggest to qualify the statement that “… CCL21trunc-driven MiniGαi recruitment virtually remained unaffected by the addition of heparin, …”.

4. Discussion: Since heparin is still widely used for anticoagulation: Do the authors assume that heparin-based anticoagulation may affect CCR7-dependent signaling?

5. I also suggest to rephrase the sentence in lines 307-311 ("Whereas …, whereas …").

6. In section 4.4, the transfection of HeLa cells with 10 µg plasmid DNA is described. Is that correct? It seems to me that this amount is higher than the values normally used (i.e., up to 5 µg).

Author Response

We thank reviewer 2 for the appreciation of the quality of our study and the helpful suggestions for improvement.

Reposnse to 1: 

We thank the reviewer for this comment. Indeed, unfractionated heparin was used for experimental procedures with an approximate weight of about 20 kDa. We have specified this in the material and method section of our revised manuscript.
A recent study has evaluated the unfractionated and low molecular weight heparin heparin in their binding affinity to CXCL11, including its clusters of basic amino acids. Notably, unfractionated and low molecular weight heparin revealed a comparable outcome in chemokine competition assays suggesting a similar function of unfractionated and low molecular weight heparin in chemokine binding (https://doi.org/10.1074/jbc.M109.082552). Additionally, initial experiments investigating the interaction of CCL21 with extracellular matrix properties mainly focused on externally applied heparin gained from porcine intestine mucosal surfaces. Subsequently, we used this unfractionated heparin for competition experiments and direct binding assays, as FPLC HiTrap Heparin HP columns are further composed of heparin from porcine intestine mucosa (https://doi.org/10.1016/S0304-4165(02)00232-5). The limited time frame of 5 days given by the editor to revise our manuscript, did not allow us to purchase different heparin variants and perform additional quantitative experiments.

Response to 2: 

We are grateful for this comment. As suggested by reviewer 2, we have modified and replaced the Figure 3b and the amended the corresponding Figure legend.

Response to 3:

We agree and have amended the statement. The sentence now reads as follows: “By contrast, CCL21trunc-driven MiniGai recruitment virtually remained unaffected by the addition of heparin and showed inhibitory effects only at very high heparin concentrations, indicating that the C-terminus of CCL21 is key for heparin binding.”

Response to 4:

Indeed, i.v. application of heparin together with antithrombin is widely used as blood thinner. We did not assess the role of heparin in vivo here. However, reference 7 describes that elimination of ECM attached heparan sulfates on LECs abrogated CCL21 presentation and efficient lymphocyte homing to lymph nodes. Whether dendritic cell homing to lymph nodes via lymphatics is affected has not been assessed. In our study, we report that heparin addition in vitro affects CCR7 signaling, but leaves CCL19-mediated CCR7 signaling intact.

Response to 5:

We agree and have modified the sentence.

Response to 6:

We thank reviewer 2 for the careful and thorough evaluation. We indeed used 10 µg of plasmid DNA for electroporation, as this amount is recommended by the manufacturer for the transfection of HeLa cells under these circumstances.

Reviewer 3 Report

By proving that heparin specifically interacts with CCL21 but not CCL19, the authors improve our understanding of the mechanism regulating the dynamics between the CCR7-CCL19/CCL21 axis and GAGs. This interaction prevents ligands from binding to CCR7 and silences signalling. The authors also demonstrate that while CCL21 variants on BBXB motifs are still able to activate G-protein-dependent CCR7 signalling, they are no longer able to interact with heparin. The findings of the assays support the idea that heparin-binding on CCL21's surface mediates this interaction and negatively regulates CCR7 signalling, and that CCL21's C-terminal residues only partially contribute to ligand-biased CCR7 signalling.

The study is interesting, however, it is unclear what the potential scientific implications are, which are not emphasised in the abstract, and in the conclusion, there is a mention of the possible importance of CCR5, which is not mentioned in the text. As a result, the abstract and conclusions should be rewritten to properly highlight the potential therapeutic implications, which are mentioned briefly in the introduction.

Author Response

We thank reviewer 3 for the careful evaluation and appraisal of our work. We summarized the major experimental findings gained from in vitro studies in the abstract and conclusion of our study. We did not dare to extend the abstract and conclusion with speculative statements about potential therapeutics implications, but have mentioned potential applications in the introduction and discussion of our study.